# Gender Differences in Urothelial Bladder Cancer: Effects of Natural Killer Lymphocyte Immunity

**DOI:** 10.3390/jcm10215163

**Published:** 2021-11-04

**Authors:** Charles T. Lutz, Lydia Livas, Steven R. Presnell, Morgan Sexton, Peng Wang

**Affiliations:** 1Department of Pathology and Laboratory Medicine, University of Kentucky, Lexington, KY 40536, USA; lydia.livas@unchealth.unc.edu (L.L.); Steven.Presnell@uky.edu (S.R.P.); morgan.t.sexton@Vanderbilt.Edu (M.S.); 2Department of Microbiology, Immunology, and Molecular Genetics, University of Kentucky, Lexington, KY 40536, USA; 3Markey Cancer Center, University of Kentucky, Lexington, KY 40536, USA; p.wang@uky.edu; 4Department of Internal Medicine, University of Kentucky, Lexington, KY 40536, USA

**Keywords:** urothelial bladder cancer, natural killer cells, androgens, immunosuppression, X chromosome, sex factors

## Abstract

Men are more likely to develop cancer than women. In fact, male predominance is one of the most consistent cancer epidemiology findings. Additionally, men have a poorer prognosis and an increased risk of secondary malignancies compared to women. These differences have been investigated in order to better understand cancer and to better treat both men and women. In this review, we discuss factors that may cause this gender difference, focusing on urothelial bladder cancer (UBC) pathogenesis. We consider physiological factors that may cause higher male cancer rates, including differences in X chromosome gene expression. We discuss how androgens may promote bladder cancer development directly by stimulating bladder urothelium and indirectly by suppressing immunity. We are particularly interested in the role of natural killer (NK) cells in anti-cancer immunity.

## 1. Introduction

The Gender Divide: “Instead of Ignoring Our Differences, We Need to Accept and Transcend Them.” Sheryl Sandberg [1].

UBC is common, with a 2.4% lifetime risk [2]. UBC is much more common in men: out of the estimated 83,750 new UBC cases in America in 2021, about 64,280 are in men and 19,450 are in women [3]. UBC also kills more men than women: in 2021 it is estimated to kill 12,260 men and 4940 women [3]. Even when controlling for increased male exposure to carcinogens, such as tobacco and industrial chemicals, men are at an increased risk [4]. Gender differences in cancer outcomes can be due to biological or social factors. For example, well-known variances in occupation, workplace exposures, and social networks could all influence cancer incidence and survival, but not be directly due to biological sex. Furthermore, physicians sometimes treat male and female patients differently. In this review, we focus on biological sex differences that may affect UBC incidence and mortality. As we discuss below, observed UBC gender differences could be related to how androgens and the androgen receptor (AR) affect UBC development and progression. UBC gender disparity also could result if androgens suppress anti-cancer immunity. Moreover, greater transcription of some X chromosome genes may protect females from UBC. 

Most UBC cases are identified early and are transurethrally resected. Adjuvant treatment often includes intravesicular infusion of cytotoxic chemotherapy or of bacillus Calmette-Guerin (BCG) immunotherapy. While non-muscle-invasive bladder cancer (NMIBC) tumors often respond to these treatments, UBC has a lifetime recurrence risk of up to 50% and requires long-term monitoring [5,6]. As a result, UBC treatment and monitoring cost more than for any other cancer [7,8]. It should be emphasized that UBC 5-year mortality rates have remained fairly consistent over the last three decades, despite advancements in treatment [2]. To help current and future patients, we must further study UBC pathogenesis. This review focuses on proposed biological reasons for high male incidence, which leads to more male deaths. What we learn from gender differences may allow us to better understand and treat UBC in both sexes. 

## 2. The Androgen Receptor in Bladder Cancer 

Encoded on the X chromosome, the androgen receptor (AR) binds to testosterone and related hormones. Testosterone stimulates cells through multiple mechanisms (Figure 1). In the classical mechanism [9], the hormone passively diffuses into the cell and binds cytoplasmic AR. Increasing potency, testosterone is reduced by 5-α reductases in the cytoplasm to dihydrotestosterone (DHT), which binds to AR with high affinity. Hormone binding changes AR conformation, which allows hormone–AR complexes to enter the nucleus and either enhance or depress transcription of multiple genes [9]. In addition, AR interacts with important cytoplasmic signaling molecules, including PI3K, Src, and Ras, to initiate MAP kinase signaling [10]. In another non-classical mechanism, ZIP9 (SLC29A9) was identified as a non-AR cell surface androgen receptor (Figure 1). Upon testosterone binding, ZIP9 increased intracellular zinc and imparted an apoptotic signal [11,12]. A putative T-lymphocyte surface androgen receptor [13,14,15] may, in retrospect, have been ZIP9. Finally, IL-6 and IL-8 inflammatory cytokines activate AR signaling in the absence of androgens (Figure 1), via ERK, Src, FAK, and STAT3 [16,17]. AR is widely expressed in both sexes, despite large differences in androgen levels. Both sexes express AR in the bladder, including urothelium, muscularis propria (detrusor muscle), and bladder neurons [18,19,20]. In castrated male rats, androgen replacement therapy significantly improved urothelial thickness and muscle fiber quantity, indicating an androgen effect on these structures [21]. 

High overall male cancer risk naturally suggests that AR and testosterone accelerate cancer initiation, progression, or metastasis. For prostate cancer (PCa), this relationship is well established [22,23,24]. In addition to PCa, many cancers are influenced by androgens. For example, castration of male rats reduced both chemically induced pancreatic tumor burden and renal cell carcinoma; testosterone administration at least partly reversed this effect [25]. However, particularly aggressive cancers often do not respond to androgen deprivation therapy (ADT). Castration-resistant PCa is a classic example. It most often arises after ADT, which selects for highly aggressive castration-resistant cancer cells that proliferate, have stem cell-like properties, and survive chemotherapy [26]. Recent work has suggested a mechanism by which this may occur—Schroeder et al. [27] found that in mice, the AR antagonist, flutamide, caused PCa cells to convert to a cancer stem cell (CSC) phenotype. The process was mediated through STAT3 signaling induced by IL-6 [27,28]. As noted above, STAT3 stimulates AR in an androgen-independent mechanism [17]. Blocking mouse IL-6 mice significantly reduced PCa stem cell numbers [27]. 

As a general principle, hormone-resistant cancers may arise from hormone-sensitive tissue. Breast cancers often are treated with tamoxifen, a selective estrogen receptor modulator that inhibits estrogen actions in breast tissue. However, some breast cancers resist hormone-based therapy. Many molecular mechanisms allow breast cancers to lose estrogen sensitivity [29]. Breast cancers that lose estrogen sensitivity typically have a more aggressive course and poorly respond to chemotherapy [30]. A notable example is triple-negative breast cancer (negative for ERBB2 (Her2/neu), estrogen receptor, and progesterone receptor protein expression), which is typically an aggressive tumor that does not respond to targeted chemotherapy agents [30]. Interestingly, triple-negative breast cancers of women often express AR and might respond to ADT or AR blockade [30]. Yet, triple-negative breast cancers that lose AR expression often have characteristics of primitive basal-like cancers with poor clinical course [31]. This suggests that AR signaling may help induce new breast cancers, but “dedifferentiated” primitive cancers may downregulate AR expression because they are driven by hormone-independent mechanisms. Similar phenomena have been observed in other cancers that become independent of the signaling molecules that characterize cells from the same tissue of origin [32,33,34]. 

Hormone independence sometimes arises early in tumor progression. Breast ductal in situ carcinomas likely are precursors of invasive carcinomas. A few of these non-invasive tumors are “triple-negative” and express other markers characteristic of highly aggressive invasive breast cancers [35,36]. This evidence suggests that steroid hormone receptors and other important drivers of breast tissue growth can be lost early in the evolution of a few tumors. Resistance to drugs that affect hormone stimulation and signature signaling pathways fits into Hanahan and Weinberg’s carcinogenesis and cancer progression paradigm [32]. Tumors are characterized by increasing independence from the growth factors that normally control cell proliferation in their tissue of origin. Independence from particular growth signals may occur early, as noted above. Due to their high mutation rate, other cancers unleash a repertoire of new growth mechanisms and acquire growth signal independence later in tumor evolution [32]. 

## 3. UBC as an Endocrine Tumor 

Growing evidence indicates that UBC, like PCa, is driven by androgens and AR signaling. Distinct from most epithelial tissues, urothelium is derived from the urogenital sinus, which also gives rise to the prostate in males [37]. We speculate that the prostate and the urothelium share properties, such as AR-dependence. Many human UBC cell lines and 13–78% of UBC tumors express AR in both men and women [38,39,40]. Moreover, stromal cells express AR in about half of UBC tumors [38]. The *AR* gene is polymorphic for exon 1 CAG repeat number, which encodes a tract of glutamine amino acid residues [41]. CAG repeat number inversely correlates with *AR* transcription and AR signal strength [41]. This suggests that the CAG-encoded polyglutamine tract interferes with *AR* transcription and possibly with AR protein function. Men with low *AR* CAG repeats are more likely to develop locally invasive or metastatic PCa [40,42]. Similarly, male UBC patients had fewer *AR* CAG repeats than control males, suggesting that AR signaling also may drive UBC [40,43]. 

In addition to spontaneously arising cancers, carcinogen-driven UBC in mice were accelerated by AR and androgens. The carcinogen, BBN (*N*-butyl-*N*-(4-hydroxybutyl)nitrosamine), more quickly induced UBC in male mice than in female mice. BBN-induced male UBC was reduced by surgical or medical ADT and an AR-blocking drug [40] or by knockout of the *AR* gene [44]. Interestingly, *AR* knockout protected both male and female mice from BBN exposure, eliminating UBC incidence and greatly reducing bladder hyperplasia [44]. Finasteride (a 5α-reductase inhibitor that prevents the conversion of testosterone to the more potent androgen, DHT; Figure 1) was marginally protective [40]. In female mice, androgen treatment increased bladder tumor incidence. The significant ADT effect and the marginal protection offered by finasteride suggest an AR signaling threshold. 5α-reductase inhibitors prevent conversion of testosterone to the more active DHT, but are associated with higher testosterone levels [45]. Therefore, mouse UBC might be driven equally by physiological male androgen signaling and by moderate androgen signaling found in the presence of 5α-reductase inhibitors. This parallels findings in human PCa—men with the lowest 10% of testosterone levels were relatively protected from PCa, but men with the highest testosterone levels had no increased risk compared to men with moderate levels [46]. Furthermore, testosterone replacement therapy does not increase PCa risk [47]. If there is an androgen threshold in UBC, we would speculate that the relatively high androgen levels in men impart a greater UBC risk. In summary, a growing body of evidence shows how AR and androgen signaling promote UBC development and progression. Based on this evidence, targeting AR is predicted to reduce UBC recurrence and progression.

Transcriptional coactivator proteins help mediate AR-dependent gene transcription and support AR protein stability [48]. The steroid hormone receptor coactivator, NCOA3 (AIB1), was overexpressed in about a third of human UBC cases, which was an independent predictor of poor progression-free survival in multivariate regression analysis [49]. Furthermore, *NCOA3* knockdown slowed human tumor xenograft growth in mice and *NCOA3* enforced expression increased tumor growth [49]. SiRNA knockdown of coactivator RNA in AR-positive UBC cell lines decreased androgen-induced proliferation [39]. AR coactivators NCOA1, NCOA2, NCOA3, CREBBP, and EP300 were expressed in 86–100% of UBCs, even though AR was present in a minority of UBCs [39,40]. The presence of these coactivators when AR was absent suggests that they cooperate in non-AR-directed transcription. 

The mechanisms by which androgens and AR control UBC are not fully understood. UM-UC-3 UBC cells selected for AR expression had higher clonogenic growth and greater migration than parental UM-UC-3 cells after androgen treatment in vitro, suggesting that AR drives UBC growth and metastasis [50]. As in prostate cancer [48], some data suggest that UBC may initially present with androgen dependence but progressive UBC may lose this dependency [40,51]. Izumi et al. [52] reported that AR, estrogen receptor-α (ERα), and ERβ nuclear staining were more frequent in normal-appearing urothelium than in UBC. In men receiving ADT, AR-negative UBC went on to recur in 12 of 28 patients (43%), but AR-positive UBC recurred in only 11 of 44 patients (23%). In multivariate analysis of men receiving ADT [52], only two factors were significant, UBC AR expression and UBC tumor number (single vs. multiple). This suggests that ADT delays or prevents UBC recurrence in AR-positive tumors (Table 1). AR expression level in normal urothelium did not correlate with tumor recurrence and tumor cell ERα and ERβ expression did not predict recurrence. In another study, Izumi et al. [53] investigated UBC recurrences in patients who were also treated for PCa (Table 1). UBC recurrence was less frequent in the ADT-treated PCa group (5-year actuarial recurrence-free survival: 76% *v* 40%; *p* < 0.001). In those men with UBC recurrence, ADT was associated with fewer recurrence episodes (5-year cumulative recurrence: 0.44 vs. 1.54; *p* < 0.001). Most patient characteristics would have been predicted to disfavor the ADT group (worse PCa disease, older age), but brachytherapy was higher in the non-ADT group and radiation is a UBC risk factor [54]. In addition, a greater percentage of the ADT group received BCG infusions [53], which prevent UBC recurrence [55,56,57]. Although intriguing, conclusions are weakened because the study was retrospective and multicenter [53]. Shiota et al. [58] followed 228 men for recurrence of bladder cancer for an average of 3.6 years; 32 of 196 men received ADT or dutasteride (a 5α-reductase inhibitor) for concomitant PCa. UBC recurred in four men with ADT or dutasteride (12.5%) and 59 men without (30.1%). Progression to muscle-invasive cancer occurred in none of the men on ADT or dutasteride and in six men who were not on these therapies (3.1%). ADT (or dutasteride) was associated with significantly decreased risk of intravesical recurrence (hazard = 0.4, *p* = 0.024). Although sample size was small and follow-up time was short, this study suggests that ADT or dutasteride may prevent UBC recurrence in high risk patients. In patients already diagnosed with UBC, retrospective studies have suggest that both overall and cancer-specific survival were superior in 5α-reductase inhibitor users [59]. ADT may protect UBC with low-risk disease, but not patients with high-risk disease [59]. This correlates with declining AR expression by more advanced UBC tumors [40,50].

Based on 13-year follow-up of the Prostate, Lung, Colorectal, and Ovarian (PLCO) database (Table 1), men who were treated with finasteride (a 5α-reductase inhibitor) at any time during the survey had lower UBC incidence (hazard = 0.634), when also controlling for age and smoking [60]. However, there was a non-significant trend toward higher UBC grades in finasteride users and there was no significant difference in UBC-caused mortality [60]. The retrospective study was limited because finasteride dose and some risk factors (such as alcohol consumption) were not known. In two retrospective insurance record reviews in Taiwan (Table 1), investigators found that 5α-reductase inhibitors did [61] or did not [62] reduce the risk of UBC recurrence. One of the studies (Table 1) found that 5α-reductase inhibitors prevented UBC deaths [62]. Interestingly, amplification of the gene that encodes 5α-reductase was found in 12% of UBC queried in the TCGA database [61]. In a systemic review, ADT or 5α-reductase inhibitors had mixed results in preventing UBC [59]. Interpretation of these studies is difficult because of the expected lower complications of urinary tract infection and hematuria in 5α-reductase inhibitor users that may have led to fewer exploratory cystoscopies and, therefore, fewer incidental UBC diagnoses. This may have decreased UBC discovery in the 5α-reductase inhibitor group. Despite the number of manuscripts showing that ADT and 5α-reductase inhibitors reduce UBC incidence or recurrence, it should be noted that most of these studies had methodological limitations [63]. 

A double-blind prospective study was conducted on subjects in the MTOPS trial [45]. Subjects were monitored for compliance, something that is impossible with retrospective studies. As expected, serum DHT levels were lower and testosterone levels were higher among the subjects treated with 5α-reductase inhibitor drug [45]. UBC incidence was not statistically different in men with and without 5α-reductase inhibitor treatment (Table 1). Thus, there is support for use of ADT in UBC, especially when the tumor expresses AR. However, the literature on 5α-reductase inhibitor treatment is inconsistent (Table 1). Literature inconsistency correlates with weak findings in experimental animals [40]. One possible interpretation of these results is that there is a threshold effect of androgen signaling in promoting UBC—the moderate amount of androgen signaling retained in 5α-reductase inhibitor users may be sufficient to promote UBC, whereas the lower level of androgen signaling under ADT may reduce UBC risk. 

Researchers have investigated a possible correlation between AR expression and increased transformation of normal bladder urothelium to UBC. In a meta-analysis, AR status in UBC correlated neither with patient gender, nor with tumor size, stage, grade, or progression [51]. In contrast, three studies showed that lack of tumor AR expression was strongly associated (odds ratio, 0.41) with more frequent UBC recurrence [51], although discordant results have been reported [40,64]. Most studies showed that AR status did not predict survival [64]. Hsu et al. [65] found that bladder-specific *AR* knockout mice developed fewer and less aggressive BBN chemical-induced UBCs. They also showed that AR acted through a p53-mediated pathway, allowing more bladder cancer cells to survive apoptosis [65]. This suggests that AR signaling increases tumorigenesis and tumor cell survival. Luna-Velez et al. reported that AR expression was significantly lower in muscle-invasive disease than in non-muscle-invasive UBC; AR expression was the lowest in the most advanced T3 and T4 disease, compared with T1 UBC [50]. On the other hand, Mir et al. [66] did not find a significant AR-related difference in time to death or rate of recurrence in a study of almost 500 patients with UBC muscle-invasive tumors. Among the patients studied, 12.9% of the tumors expressed AR and there was no significant sex difference. It is likely that AR affects UBC in complex ways that depend on cancer stage. AR signaling seems to initiate more UBCs, but we do not know whether AR signaling causes more aggressive disease. Given that many tumors grow increasingly independent of hormones and signaling pathways during cancer progression [32], it would not be surprising if androgens drive early UBCs, but not late UBCs. 

In a separate aspect of AR signaling, Miamoto’s group [67] showed that androgens increased resistance to cisplatin-based chemotherapy of AR-expressing UBC cell lines. In a complementary fashion, AR-blocking drugs increased cisplatin sensitivity [67,68]. Among the mechanisms responsible, Miamoto’s group [69] found that androgens reduced expression of an RNA- and ribosome-processing protein, BRIX1 (BXDC2), in UBC cell lines. Furthermore, AR and BRIX1 showed complementary expression patterns in UBC tissue sections [69]. Cisplatin-resistant cell lines increased BRIX1 expression [69], but BRIX1 mechanisms in UBC are not clear. Although treatment of cell lines in vitro cannot be directly translated to the clinic, these studies suggest that AR antagonists may act synergistically with cytotoxic chemotherapy in UBC patients. As mentioned above, intravesicular BCG treatment commonly follows NMIBC resection. BCG vaccination has many nonspecific effects, including nonspecific induction of “trained” immunity and lower levels of inflammatory serum proteins [70]. It is notable that the reduction of inflammatory proteins was much more pronounced in men than in women and correlated with pre-vaccine testosterone levels in males [70]. Preclinical data showed that BCG was more effective when AR was downregulated or absent, or in the presence of AR antagonist drugs [71,72]. The mechanism is possibly related to the Rab27b vesicle protein, which may allow UBC to exocytose previously ingested BCG [71]. In a retrospective analysis of human biopsy samples, both AR and Rab27b protein expression correlated with higher UBC recurrence rates [71]. 

UBC incidence and the total number of UBC-related deaths are much higher in males than in females [3], but women suffer higher stage-for-stage progression and mortality than do men [73,74,75]. Sadly, some of the greater mortality in women may be related to delayed treatment or different treatment opportunities offered to females [74]. For example, urological investigation of hematuria is delayed in female patients compared with male patients. As an exception to that general finding, women may have a better cancer-specific mortality than men following neoadjuvant chemotherapy plus radical cystectomy [76]. This conclusion, which was based on two small studies [76], requires confirmation. However, women have worse outcomes following transurethral resection of NMIBC and post-surgery chemotherapy [74]. In early stage Ta tumors, women are more likely to have the GS1 UBC subtype. These tumors have high rates of proliferation, mutations, genetic instability, and loss of chromosome 9 or deletion of 9q [77]. These deletions remove TSC1, a negative regulator of the mechanistic target of rapamycin (mTOR). Consequently, GS1 NMIBCs have gene expression changes that suggest altered metabolism, including elevated glycolysis [77]. Consistent with overall worse outcomes for women, multivariate analysis suggested that female sex was a risk factor for recurrence after BCG therapy [74,78], despite the well-documented superior responses by women to a variety of vaccines [79,80,81,82,83,84]. de Jong and colleagues [85] found that, compared with men, women were more likely to have the aggressive basal/squamous subtype, whereas men were more likely to have luminal papillary and neuroendocrine-like subtypes. It may be significant that advanced UBC in men had higher androgen response activity across all luminal subtypes, compared to advanced UBC in women [85]. These findings support the concept that androgens help initiate UBC. However, some UBCs progress to a more aggressive form, becoming androgen-independent through AR loss or other means. One molecular consequence of AR loss in advanced UBC is that AR suppresses transcription of *CD44* [86], which encodes a cancer stem cell marker that mediates UBC aggression [87]. AR mRNA negatively correlated with CD44 mRNA in UBC, both before and after chemotherapy [86]. Due to lower androgen levels in women, it should not be surprising that the smaller numbers of UBC female patients are enriched for more aggressive androgen-independent cancers. In contrast to androgens, the role of estrogens in UBC is less clear. Estrogens appear to suppress UBC development but may promote UBC progression [74]. Treatment of mice with the selective estrogen receptor modulator, tamoxifen, greatly reduced bladder tumor formation and muscle-invasive tumors in female mice fed the BBN carcinogen [88]. Hsu et al. [89,90] reported that female mice lacking ERβ were *less* susceptible to carcinogen-induced UBC, whereas female mice lacking ERα were *more* susceptible. These apparently discordant results suggest that there are distinct roles for ERαα and ERββ homodimers, in addition to ERαβ heterodimers. 

## 4. Androgenic Immune Suppression

Gender-based immune disparity is well-documented [81,91,92]. Men suffer more infectious disease complications [93,94] and cancers [3,95,96,97] than do women, including at elderly ages. Women generally respond better to vaccines, although exceptions exist [79,80,81,82,83,84]. Autoimmune disease can be attributed to excess immunity and women are more susceptible to the most common autoimmune diseases [83]. Sex differences are observed in children as well. Infant boys are significantly more susceptible to infections than infant girls, a difference that is attributed to the androgen surge that boys experience at birth [83,98,99,100]. In addition to androgens, male vs. female immune response differences might be linked to estrogens, progesterones, X-linked genes, and socioeconomic factors. In the current pandemic, male gender has been found to be a strong risk factor for COVID-19 disease and death, whether or not overall SARS-CoV-2 infection rate is higher in men than in women [101,102]. 

Early hematopoietic precursor cells express AR RNA in mouse bone marrow and in human bone marrow and cord blood [103]. However, some differentiated hematopoietic cells lose AR expression [103]. Using immunohistochemistry, Mantalaris and colleagues [104] detected AR protein in several bone marrow elements, including stromal cells, endothelial cells, Mφ, and other myeloid cells (although not eosinophils). AR expression was affected neither by sex nor by age (range 1–92 years). However, Mantalaris et al. [104] did not detect AR protein in bone marrow lymphocytes.

## 5. Androgens and Innate Immunity—Myeloid Cells

Myeloid cells are pivotal in anti-cancer immunity [32,105,106,107]. Depending on the local cytokine and cellular microenvironment, myeloid cells promote tumor angiogenesis and growth and suppress immunity. Alternatively, myeloid cells present tumor antigens to T cells and cooperate with NK cells to eliminate cancer cells [32,105,106,107,108,109,110,111,112,113,114,115]. Developing myelocytes respond to androgens, but specific myeloid subsets become androgen-independent at distinct differentiation stages. The common myeloid progenitor is stated to express AR RNA [25]. The common dendritic cell and the granulocyte–macrophage progenitor do not express AR, but mature myeloid cells do, including macrophages (Mφ), monocytes, neutrophils (both band cells and segmented), and mature mast cells [25,104].

Among the myeloid cells, Mφ are particularly important because they regulate tissue homeostasis and produce several proinflammatory cytokines [108,114,116,117,118,119,120,121]. Myeloid-specific *AR* gene deletion and AR-blocking enzalutamide reduced monocyte precursors in the mouse bone marrow [122]. *AR* knockout also reduced the proportion of classical monocytes [122], which are thought to enter tumor and inflammatory sites and develop into macrophages or dendritic cells. Gonadectomy elevated pro-inflammatory responses by increasing expression of toll-like receptor 4 (TLR4) by male murine Mφ [123]. Some results, however, suggest that testosterone promotes specific proinflammatory cytokines in certain contexts [124,125,126]. Thus, testosterone and AR signaling may differentially affect Mφ depending upon the Mφ source, the type of stimulus, and the cytokine secreted. Mφ cells involved in wound healing express AR. Wound healing is slower in males than in females, but healing has been shown to be accelerated by an AR antagonist, flutamide, and by myeloid-specific *AR* gene deletion [124,125]. Importantly, wound healing correlated negatively with testosterone levels in elderly men [124].

Neutrophils, the most abundant blood leukocyte, respond rapidly to bacterial infections and many cancers [127]. Androgens promote neutrophil differentiation—AR-deficient mice, androgen insensitive mice, and AR blocker-treated patients are neutropenic [128,129]. Following treatment with stanozolol, a testosterone analog, female mice increased neutrophil maturation rate [130]. However, the mechanisms by which AR signaling affects neutrophil function have not been fully elucidated [131]. Some authors have shown that androgens suppress neutrophil production of proinflammatory cytokines and instead promote anti-inflammatory IL-10 production [132]. These data suggest that androgens promote neutrophil differentiation, but dampen neutrophil inflammatory actions.

Dendritic cells (DCs) present antigens to T cells and affect NK cell function [109,110,111,112,113,133,134]. DCs synthesize IL-15 and present it on their cell surface to stimulate NK cells and memory T cells to mature, divide, and survive [110,135,136,137]. DCs stimulate NK cells to synthesize granzyme B and become cytotoxic [110,135,137,138]. Only a few studies have addressed possible direct effects of AR and androgens on DCs, but, in general, they seem to depress DC immune function [25]. This is controversial because some investigators have shown that DCs do not express AR [139]. DCs isolated from male mice after brain LCMV infection were less activated than in female mice [140]. Male sex reduced expression of DC MHCII and CD86 [140]. These molecules, respectively, present antigens at the cell surface to CD4 T cells [141] and co-stimulate T cell CD28, which promotes T-cell differentiation and survival [142]. Castration increased expression of MHCII, CD86, and other co-stimulatory molecules [143]. 

## 6. Androgens and Innate Immunity—NK Cells

NK cells kill both virus-infected cells and cancer cells, including metastases, without antigen specificity or prior immunization [115,144,145,146,147,148,149,150]. Comprising 5–15% of blood lymphocytes in healthy people, NK lymphocytes are defined by CD56, CD16, or NKp46 (CD335) expression in the absence of the CD3 T-cell receptor [144,145,146]. NK cell activation is regulated by multiple receptors [144,145,146]. Cell surface MHC class I molecules (MHCI, termed HLA class I in humans) present antigenic peptides to CD8 cytolytic T lymphocytes, which then divide, secrete IFN-γ and other cytokines, and kill antigen^+^ cells [141,151]. In addition to stimulating CD8 T cells, MHCI molecules strongly modulate NK cell responses by engaging NK cell KIR, NKG2A, and LILRB1 inhibitory receptors [144,146,152,153]. NK cells also express many stimulatory receptors [144,145,146]. For example, NKG2D binds to stress-activated ligands that are preferentially expressed on tumor cells and virus-infected cells [144,146,148,154]. When confronting stress-activated ligand-positive tumor cells, antibody blockade of NKG2D prevents robust NK cell activation [154]. 

AR may affect NK cell development. Acyline-mediated chemical castration (ADT) for two weeks increased circulating NK cells in men [155]. Testosterone replacement prevented these changes. After 4 weeks of recovery from ADT, NK cell number returned to normal levels; CXCR1 and NKG2D expression did not significantly change [155]. These findings indicate that androgens control human NK cell numbers. In the elderly, the immature CD56^bright^ to mature CD56^dim^ NK cell ratio was significantly higher in women than in men [156]. Female CD56^dim^ NK cells had higher cytotoxic granule exocytosis in response to K562 tumor cells and higher IFN-γ made in response to cross-linking of NKp46. Plasma IL-15, a cytokine required for NK cell development and survival, did not differ by gender [156]. Therefore, although the mechanisms of the gender differences were not identified, the results suggest that testosterone suppresses NK cell activity in elderly humans. We did not detect AR mRNA in either CD56^bright^ or CD56^dim^ peripheral blood NK cells (data not shown); therefore, the effects of testosterone on mature NK cells likely are indirect. We further investigated whether androgens affected NK cell activity in the context of cancer treatment. After obtaining informed consent, five men were studied before and after ADT for metastatic PCa. The study was IRB-approved and consistent with the Helsinki Declaration. Peripheral blood was removed by venipuncture and the ability of NK cells in a mononuclear cell preparation to produce a chemokine (MIP-1β) was compared pre- and post-ADT. Both IL-15 and IL-12/IL-18 stimulated significantly higher MIP-1β responses post-ADT, with the IL-2-stimulated responses trending higher (Figure 2). Several stimuli failed to significantly increase IFN-γ or cytotoxic responses post-ADT. This shows that NK cells from men receiving ADT were not globally activated, but selectively produced MIP-1β in response to IL-15 and IL-12/IL-18 cytokine stimuli. Although many changes take place with ADT initiation, including a reduction of PCa mass, our results are consistent with the hypothesis that androgens suppress NK cell MIP-1β responses to cytokines and that ADT relieves the suppression.

## 7. Androgens and Adaptive Immunity—B Cells

B cells both activate and carry out adaptive immunity. Firstly, B cells stimulate a special class of CD4 T follicular helper cells in the germinal center [157]. Later in the immune response, B cells develop into plasma cells that secrete antibody, which protects the host by several mechanisms [158]. In general, women produce relatively more antibody after vaccination [80,81]. Low testosterone levels in men predict more B cells and higher response to vaccines and to infection [80,83]. Research suggests that mature B cells do not express AR; therefore these effects may be exerted by other leukocytes or on B-cell precursors when the AR is expressed [159,160,161]. Several lines of evidence show that AR signaling impedes B-cell development. [159,162,163,164]. Experiments with chimeric mice that expressed *AR* exclusively on either stromal cells or lymphoid cells in the bone marrow showed that stromal cell *AR* is essential to inhibit the B-cell lymphopoiesis typically observed with androgen treatment [164]. This suggests that the B-cell response to androgens is mediated by stromal cells. In contrast to these findings, B-cell-specific *AR* knockouts have been shown to elevate B-cell lymphopoiesis, suggesting that mouse B-cell precursors are direct androgen targets [164]. The effect of the general *AR* knockout was more pronounced than the B-cell-specific *AR* knockout, suggesting that both B-cell and stromal *AR* suppress B-cell development. Further studies have shown that DHT causes stromal cells to produce transforming growth factor-β, an anti-inflammatory cytokine that dampens immune responses. Relevant to this discussion, transforming growth factor β suppresses IL-7 production, a cytokine that is required for B-cell proliferation and differentiation [165]. 

## 8. Androgens and Adaptive Immunity—T Cells

In comparison with B cells, T-cell response to androgens is more sustained and direct [84]. AR is highly expressed in lymphoid precursor cells and in supporting cells—marrow stromal cells and thymic epithelial cells [166]. It was established more than a century ago that castration causes thymic hypertrophy, which suggests that androgens regulate T-cell development [91,167,168,169,170]. Although AR has been detected in thymocytes, experiments with chimeric mice showed that thymic involution required AR expression in stromal cells, but not in bone-marrow-derived T lymphocyte precursors [166,169]. In a key molecular step, androgens reduce thymic epithelial expression of δ-like 4, a notch ligand that is required for T-cell maturation [91,171]. 

Mature T cells appear to express both classic cytoplasmic AR and a plasma membrane androgen receptor [13,14,15,161,172,173,174]. Exogenous androgen treatment skews mouse T-cell activation, proliferation, and differentiation, and inhibits T-cell-dependent antibody production [10,25,80,83]. These effects are likely due to a combination of intrinsic T-cell effects [172] and depression of antigen-presenting cell MHC and costimulatory molecule expression or cytokine production [123,124,125,126,140,143]. Thus, androgens affect both T-lymphocyte development and mature T cells. 

The study of androgen regulation is complicated by the many classes of T cells, each with unique functions. CD4 T-helper 1 (Th1) cells are proinflammatory and produce cytokines that stimulate cell-mediated and innate responses, which help prevent tumors and clear intracellular bacteria and viruses [175]. Kissick et al. [172] showed that testosterone reduced CD4 Th1 differentiation by upregulating protein tyrosine phosphatase, non-receptor type 1 (Ptpn1), which negatively regulates many cellular processes. T cells expressed less Ptpn1 in PCa patients undergoing ADT than in control PCa patients [172] and (Haydn T. Kissick, personal communication 12 April 2019). CD4 T-helper 2 (Th2) cells control humoral immunity and the clearance of extracellular pathogens [175]. As with Th1 cells, androgens diminish Th2 responses [176]. Th2 cells also quell inflammatory immune responses by secreting anti-inflammatory cytokines [175,177]. Experimental autoimmune encephalomyelitis is a commonly used model for multiple sclerosis, a demyelinating autoimmune disease. In experimental autoimmune encephalomyelitis cell culture and mouse models, androgen treatment increased IL-10 production and decreased demyelinating disease in female mice [178]. The underlying mechanisms are not known, but IL-10 generally seems to protect against autoimmune disorders [177]. Androgens stimulated mast cells, which activated innate lymphoid cell type 2 (ILC-2) cells, which in turn activated a Th2-like response [179,180]. Interestingly, testosterone, Fc receptor cross-linking, and Mycobacterium tuberculosis all induce IL-33 production in male, but not in female, bone-marrow-derived mast cells [179]. In specific inflammatory diseases, IL-33-driven Th2 responses ameliorate disease pathology [181]. 

Regulatory T cells (T_reg_) modulate the immune response by reducing inflammation and by modulating response to self-antigens [182,183]. Regulatory T cells put the brakes on a variety of cells, including NK cells [184,185]. Androgens increase T_reg_ cells in vivo and in vitro. Waleki et al. showed that this modulation was related to androgen-dependent acetylation of histone H4 at the *FOXP3* locus, a gene that is needed for T_reg_ cells to differentiate in the thymus and to function in the peripheral tissue [174,186]. Androgen modulation of FoxP3 expression is one of many mechanisms by which androgens suppress immunity. 

## 9. Chromosomal Effects on UBC

In healthy females, each cell randomly silences one X chromosome at an early stage of embryogenesis [187]. However, X chromosome silencing is incomplete and about 15–25% of these genes are transcribed to at least the 10% level on the active X, and a few at nearly 100% [188,189]. Some genes always escape complete inactivation, and some vary between XX females. Escape from complete silencing also differs by tissue and stage of development and increases with entry into the cell cycle [190,191]. Thus, compared to males, females express significantly more mRNA from certain X-linked genes. UBC and other solid tumor incidence are *increased* in patients with Turner syndrome, which is characterized by XO, a single sex chromosome [192,193]. Similarly, solid tumor incidence is *decreased* in Klinefelter syndrome patients, who carry at least 2 X chromosomes and 1 Y chromosome, have small testicle size, and reduced testosterone [192]. As in females, X chromosome inactivation is incomplete on supernumerary X chromosomes in Klinefelter males [194]. Both Turner syndrome and Klinefelter syndrome patients have altered sex hormone levels, so sex hormones, X chromosome number, or both might affect cancer risk. To separate biological sex and associated hormone expression from sex chromosome composition, investigators have created “four core genotype” mice [195]. These animals have the sex-determining *SRY* gene deleted on the Y chromosome and inserted into an autosome, allowing biological sex (and sex hormones) to be separated from X and Y chromosome composition. Using the “four core genotype” mice, Kaneko and Li [196] studied UBC responses to the BBN carcinogen. XY male mice were most susceptible, followed in order by XX males, XY females, and XX females. Survival was significantly different in each group. Thus, both biological sex and sex chromosome composition contributed to UBC risk in mice. Investigating the mechanism involved, Kaneko and Li [196] showed that the X-linked lysine demethylase 6A (*KDM6A*) gene is expressed more highly in the urothelium of XX males and females than in the urothelium of XY males and females [196]. This is because females express *KDM6A* from both X chromosomes, escaping silencing [197,198]. Mechanistically, KDM6A reverses the action of EZH2 methylation, which inhibits the transcription of multiple genes [199,200]. These mouse findings are relevant to human disease. UBC from female patients had more KDM6A expression than did UBC from male patients [77,196]. Furthermore, low tumor KDM6A expression was associated with higher UBC tumor stage in women, but not in men. *KDM6A* mutations and low mRNA levels correlated with poor disease-free survival in women, but not in men [196]. *KDM6A* is mutated in several cancers, but the highest rates of mutation (20–29%) are found in UBC, with most of the genetic alterations being pathogenic nonsense, frameshift, or splice site mutations [199,201,202,203]. An exciting finding is that loss of *KDM6A* in a UBC xenograft model predisposed tumor cells to epigenetic therapy, which diminished in vivo tumor growth and increased natural killer cell attack [203]. In addition to *KDM6A*, Dunford et al. [204] found five other X-linked genes (*ATRX, CNKSR2, DDX3X, KDM5C,* and *MAGEC3*) with loss of function mutations or copy number changes that were more frequent in men than in women in many cancer types. This sex-linked association was highly significant, as it was found in 6 out of 783 X chromosome genes but in zero of 18,055 genes from autosome chromosomes or pseudoautosomal X chromosome regions [204]. Although we do not fully understand how most of these X chromosome genes contribute to cancer, their incomplete allelic inactivation may protect women.

The Y chromosome also affects cancer risk, but functions differently [205]. From an analysis of 1153 elderly men, partial loss of the Y chromosome in peripheral blood was associated with increased non-hematological cancer mortality. Men who had lost the Y chromosome had a median survival time of 5.5 years shorter than patients who had an intact Y chromosome [205]. More relevant to this discussion, the Y chromosome is uniformly missing from many male UBC cases [206]. In other cancers with a defective KDM6A allele on the X chromosome, the Y chromosome may be present in the cell, but may severely limit gene expression [202]. These findings suggest the possibility of one or more Y-linked tumor suppressor genes. One candidate for the Y chromosome tumor suppressor gene is *KDM6C* (also known as *UTY*), which is a paralog of *KDM6A*, mentioned above. Ahn et al. [207] found that 23% of male UBC cases had deleted *KDM6C*; the *KDM6C* deletion rate was 67% in UBC that also had an X chromosome *KDM6A* mutation. Other investigators found *KDM6C* copy number loss or Y chromosome loss in 12–42% of male UBC cases and this was more common in UBCs that carried *KDM6A* mutations [77,199]. Thus, the X chromosome *KDM6A* and the Y chromosome *KDM6C* each may suppress tumor growth and partially compensate for each other in males.

## 10. Sex Differences Not Directly Attributed to Hormones or Sex Chromosomes

Some sex-related differences do not appear to be directly related to sex hormone differences or sex chromosomes. For example, chromosome 3 *VGLL3* shows female-biased expression in skin, salivary glands, and in monocytes [208]. Keratinocytes from female subjects expressed more VGLL3 RNA, whether there was estradiol, testosterone, or no sex hormone in culture [208]. *VGLL3* is of interest as it may act as a tumor suppressor gene in epithelial ovarian cancer [209], but *VGLL3* expression is a marker for poor outcome in stomach adenocarcinoma and in PCa [210,211]. Therefore, *VGLL3* is differentially expressed by sex in some tissues and alters tumor outcome. In another example, Sabag et al. [212] found that hemizygous deletion of the chromosome 10 *EGR2* (Krox20) gene, whether in whole animals or in myeloid cells, increased bone resorption by osteoclasts in female mice, but not in male mice. The phenomenon continued to be manifested in preosteoclast cell lines cultured several days in vitro in the absence of any sex hormones. However, further investigation [212] showed that the sex differences were manifested only in preosteoclast cells that had been derived from post-pubertal mice (8 weeks old) but not harvested from mice before puberty (4 weeks old). These and other data [213,214] suggest that sex hormones cause epigenetic differences in the genomes of male and female animals, which persist long after sex hormones are removed. It is relevant to note that most genes that show sex-specific expression are driven by distinct sets of transcription factors in male and female tissues, even when the transcription factor levels do not vary by sex [215]. This strongly suggests that the sex differences establish distinct epigenetic landscapes, which persist after direct sex hormone action. Many of the differentially expressed sex-specific proteins are involved in methylation and presumably control epigenetic gene expression [215]. 

## 11. UBC and Natural Killer Cells

NK cells are the most common leukocyte in UBC tumors [216]. Using gene expression data to deduce immune cell infiltration, several groups have reported that the presence of NK cells predicts better UBC outcomes [203,217,218,219,220]. Importantly, the level of CD56^bright^ NK cells correlates with both cancer-specific survival and overall survival [216]. These data are consistent with the idea that NK cells slow UBC progression. This idea is reinforced by findings that epigenetic therapy of *KDM6A*-deficient UBCs caused NK cell attack and tumor regression in a preclinical model [203]. As mentioned above, NMIBC is often treated with intravesical BCG instillation [55,56,57]. Now used for more than 40 years, BCG intravesicular infusion is the longest continuously used cancer immunotherapy [55,56,57]. NK cells are among many immune and non-immune cells that are required for BCG immunotherapy success [55,221,222,223,224,225]. In other settings, BCG vaccination induces an elevated “trained immunity” against multiple pathogens, including elevated in vitro NK cell responses against non-cross-reactive microbes [226]. Surprisingly, intravesicular infusion of cytokine-activated healthy donor human NK cells were reported to eliminate orthotopic human UBC mouse xenografts [225]. If this study is confirmed and expanded to human patients, it would suggest the potential for NK cell-based therapy. 

UBC patient NK cells differ from those of healthy subjects. NK cells from UBC patients expressed low levels of L-selectin (CD62L), which allows lymphocyte circulation through lymph nodes, and low levels of stimulatory NKp30, NKp44, and NKp80 natural cytotoxicity receptors, which are crucial for target cell killing and other NK cell activities [146,225,227]. IL-2 boosts NK cell responses in cancer patients [228]. Following subcutaneous IL-2 injection, circulating NK cell number increases [229]. Importantly, some melanomas and kidney cancers respond to IL-2 injections [230]. However, IL-2 is a strong activator of regulatory T cells, which dampen immune responses by both T cells and NK cells [183,228,231,232]. IL-15 is an alternative cytokine because it signals NK cells and memory T cells though the same receptor β and γ chains as IL-2, but it renders effector cells refractory to regulatory T-cell suppression [233,234,235]. Using NK cells as a therapeutic agent, healthy donor NK cells were stimulated in vitro and then infused into hematopoietic stem cell transplant patients, with NK cell responses boosted by IL-2 injection. This combined treatment prevented acute myeloid leukemia relapse [236]. 

CSCs drive cancer progression and long-term proliferation by maintaining an undifferentiated state and replacing partially differentiated cancer cells [32,237,238,239,240,241]. CSCs resist radiation and many chemotherapeutic drugs, and account for metastases and post-treatment tumor relapse [32,237,238]. NK cells preferentially target CSCs [225,242,243,244,245], which suggests a possible therapeutic approach. UBC-derived NK cells killed the RT4 UBC cell line much more effectively after blocking MHCI [216], consistent with functional MHCI-specific inhibitory receptors on UBC tumor-infiltrating NK cells. Although CSCs may have relatively low or high MHCI expression, CSCs are NK susceptible because they express abundant stimulatory ligands [242,243,244,245]. The NK-stimulating ligands expressed by CSC include NKG2D ligands: MHC class I chain-related protein A and B (MICA and MICB) and ULBP1. Killing of CSCs by NK cell sis primarily controlled through the interaction of the NKG2D receptor and its ligands [225,245]. In mouse models, NK cells significantly decreased UBC CSC number [225]. These data suggest that NK cells kill CSC and thereby prevent bladder cancer recurrence. 

## 12. Conclusions

In most non-reproductive tissues, men are more likely than women to develop cancer and die of disease, and UBC is an outstanding example. Gender disparity has many potential causes, including androgen-induced immunosuppression, androgen-induced urothelial and neoplastic proliferation, greater male exposure to carcinogens, and the protective effects of X chromosome genes in females. We predict that better understanding the gender differences will to lead to more effective treatments in both men and in women. Suppression of androgens [40] may reduce NMIBC recurrence, progression, and mortality in men, and possibly in women. Knowledge that KDM6A and KDM6C are under-expressed in UBC suggests that reducing EZH2 activity will effectively treat KDM6A/KDM6C-defective UBC in both sexes [199,200]. Boosting immunity, especially NK cell immunity, may lead to better outcomes in both sexes, especially in men. Although this review largely focused on biological sex, social factors also are important: women often receive delayed diagnostic testing and disparate treatment options. Finally, when urologists recognize that females and males are sometimes treated differently [74], outcomes may improve. We believe that a better understanding of gender-specific disparities in UBC, at the genetic, molecular, cellular, clinical, and cultural levels, will lead to more safe and effective treatments for both male and female UBC patients. More research is urgently needed. 

## Figures and Tables

**Figure 1 jcm-10-05163-f001:**
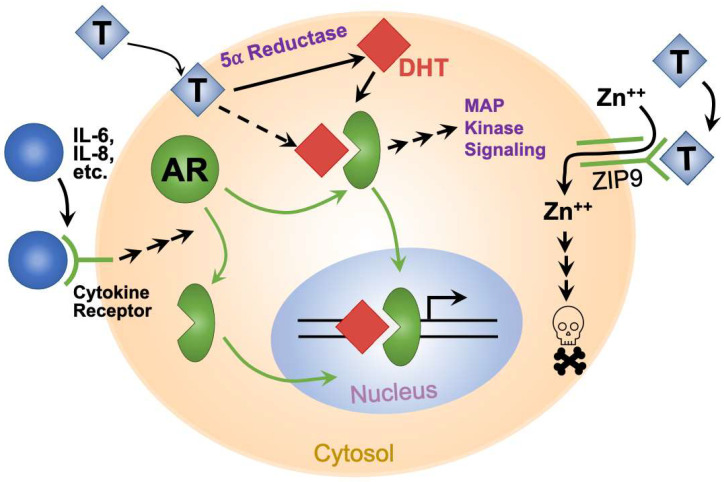
Schematic representation of androgen and AR signaling mechanisms in cells.

**Figure 2 jcm-10-05163-f002:**
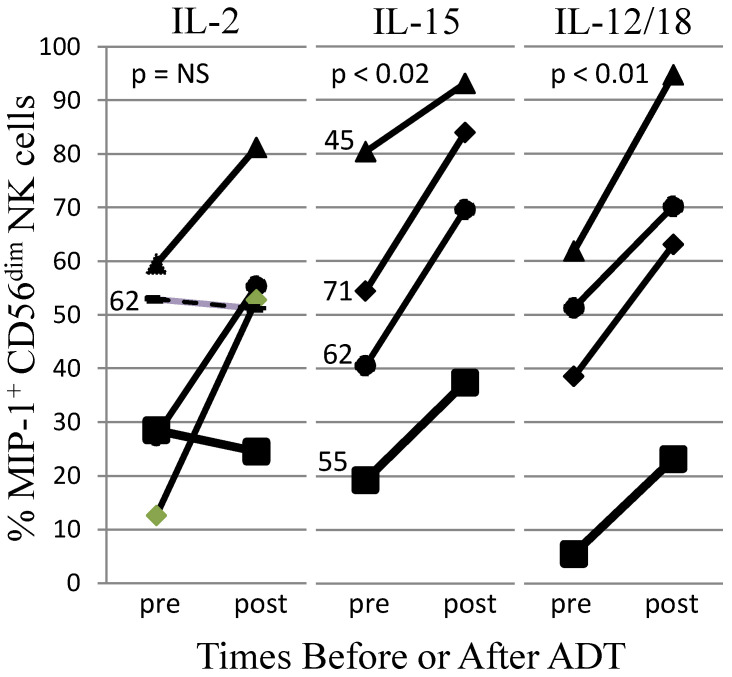
Blood from five patients was studied before and 92–96 days (or 51 days in one patient shown as a dashed line in IL-2 panel) after the start of ADT, which included Lupron^®^, with or without additional ADT drugs. In all cases, short-term Casodex^®^ pre-treatment was used to prevent a Lupron-associated testosterone flare. Peripheral blood mononuclear cells were stimulated for 20 h with IL-2 (200 U/mL), IL-15 (100 ng/mL), or IL-12/IL-18 (10/100 ng/mL), as indicated above each panel. The cells were harvested and NK cell intracellular MIP-1β was measured by flow cytometry, using standard methods [156]. Shown are responses by mature CD56^dim^ NK cells. Each symbol represents responses by the same patient and patient age is indicated in the panels. Significance was assessed by paired student’s *t*-test. NS = not significant.

**Table 1 jcm-10-05163-t001:** Effects of Androgen-Based Therapy on UBC Recurrence, Progression, and Mortality.

Treatment	Effect on UBC	Support *	Ref.
ADT	↓ recurrence ^#^	Yes	[52]
ADT	↓ recurrence	Yes	[53]
ADT or 5α-R ^$^	↓ recurrence, progression	Yes	[58]
5α-R	↓ incidence, ↔ mortality	Yes/No	[60]
5α-R	↓ recurrence	Yes	[61]
5α-R	↔ recurrence, ↓ deaths	No/Yes	[62]
5α-R	↔ incidence	No	[45]

* Support for the hypothesis that UBCs respond drug treatment. ^#^ Recurrence was less in men with AR^+^ UBC ^$^ 5α-R, 5α-reductase inhibitor drug.

## Data Availability

Data will be made available upon request to the first author.

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
