# Peer review of "Gender Differences in Urothelial Bladder Cancer: Effects of Natural Killer Lymphocyte Immunity"

_jcm, 2021, doi:10.3390/jcm10215163_

Round 1

Reviewer 1 Report

The article has been consistently improved. I recommend its publication. I have no further comments. 

Author Response

We thank Reviewer #1.

Reviewer 2 Report

This paper is well written and presents interesting hypotheses on the role of NK lymphocyte immunity in bladder cancer, with therapeutical implications.  It needs, before publication, a number of changes, which are outlined below:

  1. Abstract: the last sentence of the abstract needs to be rephrased in a more scientifically sound way "We are particularly interested in natural killer cell (NK) because they are important, but often overlooked anti-cancer lymphocytes". This sentence does not seem to be acceptable as the abstract conclusion. The authors should rephrase the sentence, avoiding using "often overlooked", which seems a too generic assumption.
  2. This paper presents a large number of references, but I think it should include also "Spotlight on gender-specific disparities in bladder cancer", Mancini et al., Urologia J 2019, because it highlights some evident gender stereotypes in bladder cancer management, which are not mentioned in the author's paper. It would be very interesting for the readers of JCM to read this reference, to increase the reader's awareness on gender-related discrepancies in bladder cancer management. I would definitively include the Urologia J paper in the references cited in the Introduction, and also in Chapter 3 (UBC as an endocrine tumor), particularly on page 7. Finally, I would add it to the Conclusions.
  3. I would not use, throughout the paper, the definition "superficial" when referring to bladder cancer, since this definition is obsolete. Use the current acceptable definition of  "Non muscle-invasive".
  4. At the end of the Introduction, there is the sentence " What we learn from gender differences may allow us to better understand and treat UBC in both sexes". Do the paper's Conclusions respond to this hypothesis? I think the Conclusions need a little expansion to respond to this last line of the Introduction.
  5.  I think that the sentences in the paper should be put in the impersonal form, like for example " page 4:  "We do not completely understand how androgens and AR control UBC". It would be better to write " The machanism by which abdrogens and AR control UBC is poorly or not completely understood".
  6. In the Conclusions, it is said that "Gender disparity has many potential causes, androgen-induced immunosuppression, etc. etc". I think you need to add cultural differences, gender-related sterotypes, less frequent referrals to the Urologists of women as compared to men (cit. Mancinie t al., Urologia J,  2019). The complexity of the gender-related disparities in UBC needs to be better highlighted, especially in the Conclusions of the paper.

Author Response

We thank Reviewer #2 for their assessment that the manuscript is well-written. We have complied with all requests, as discussed in the specific comments below. Changes made since the last review are shown by the Microsoft Word “markup” function.

  1. The Abstract sentence has been rephrased.
  2. Mancini et al has been cited five times (#75). We directly or indirectly refer to this citation in the Introduction, Section 3, and the Conclusion. We thank Reviewer #2 for pointing out this citation.
  3. We replaced the term “superficial.”
  4. We thank Reviewer #2 for pointing out that we should tie together the Introduction and Conclusion.
  5. As requested, several sentences have been transformed from the active voice to the impersonal passive voice.
  6. The gender disparity in female UBC care was highlighted in the Conclusion.

Round 2

Reviewer 2 Report

The manuscript has been improved in several parts, and it is now more complete and  scientifically sound.  The Conclusions need to include a link to future reaserch streategies and to give a take home message (always a good thing for a review). So they  need to be rephrased to be more accurate and open on future perspectives.  I insert below an example of how this could be done:

Conclusions:

In most non-reproductive tissues, available data shows that men are more likely than women to develop cancer, including UBC, and die for it. However, when looking at UBC in different stages of disease and clinical settings, women can show worse prognosis and reduced treatment opportunities. Several gender-related issues could be potential causes of this phenomenon, such as cultural differences,  gender-discrepancies in exposure to carcinogenes, hormones-induced urothelial proliferation and carcinogenesis, androgen-linked immunosuppression, like suppression of NK response, (a promising target for immunotherapy in UBC), and the protective role of X chromosome-linked genes in females. Moreover, gender-related differences in UBC treatment opportunities cannot be underestimated. It is our belief that better understanding of gender-specific disparities in UBC, at the genetic, molecular, cellular, clinical  and cultural level will lead to more effective and oncologically safe treatments for both male and females patients with bladder cancer in the next future. More research is needed in this most urgent field.

Author Response

As directed by Reviewer #2, the Conclusion was revised, while retaining the same writing style of the rest of the manuscript. It should be noted that the review was primarily directed at biological sex factors and not social and cultural factors. Therefore, it would be inappropriate to over-emphasize one aspect of cultural differences (apparently slower diagnosis and worse treatment of women), but ignore other gender factors, some of which disadvantage men (weaker social networks, greater reluctance to seek help, etc).

This manuscript is a resubmission of an earlier submission. The following is a list of the peer review reports and author responses from that submission.

Round 1

Reviewer 1 Report

This is the first manuscript I have been asked to review where it is already formatted for publication. Therefore my comments will be aimed at improving the quality where possible.

Introduction.

To describe UBC as more deadly in men should be reworded: the rate of death is higher in women though it kills more men in total. This is an important point given that the discussion is framed on this erroneous description of UBC as more “deadly” in men, whereas it is actually more common and less deadly.

Consider using actual numbers from prior years instead of estimated numbers for this time of comparison.

Men are at increased risk for diagnosis, but again the death rate per diagnosis is higher in women.

References should be provided when stating a 50-70% recurrence rate; 70% is too high according to most reliable studies.

The section on prostate cancer could be more succinct and avoid certain details. For instance, a 2014 paper on the potential role of CSCs is neither recent nor seminal work in the large field of mechanism of castration resistance. The term castration resistant is preferred to hormone-resistant for prostate cancer.

“Similar phenomena have been observed in other cancers that become independent of the signaling pathways that characterize cells from the same tissue of origin.” When making statements like this, it is appropriate to include references.

Section 3.

While the authors suggest the common origin from the urogenital sinus suggests a reason for AR-dependence, this is speculation. Further, the prostate is considered to be mesoderm origin, which the urothelium is endodermal.

In the discussion of the role of AR in patient bladder cancer, the summary statement of methodological flaws is overly harsh as while each study has limitations, they are not necessarily flawed. Similarly, the MTOPS study is not the most definitive study as indicated in the text; it is concerning that this is even mentioned given the obvious lack of power in this study. For a formalized assessment of bias of these articles and several not included, as well as a more balanced presentation the corrections should aspire to present, the authors should refer to PMID: 33132108.

A recent article also nicely summarizes the data on AR expression in bladder tumors: PMID: 32676741 which present data on clinical outcomes related to AR bladder cancer expression.

Section 5

Similarly, PCa patients treated with AR blockers may display neutropenia [128]. This is an off-target effect related to flutamide and not AR antagonists. Multiple other AR antagonists do not cause neutropenia.

Section 6

It is highly unusual for a review article to switch in the middle to cite large amounts of unpublished data. The switch from narrative to first person plural also does not read well. Considering omitting this, as the review does not suffer from being to brief.

Section 7,8

Like much of the article, these sections are very detailed. The density of material is high, making it difficult to read. The authors may consider to clearly explain a few concepts, while referring others with less explanation to make it more readable.

Section 11

It might be better to rephrase the last phrase of the first paragraph since this is highly speculative. Perhaps something along the lines: “this suggests the potential for NK-cell based therapy”; it is not just a confirmatory study needed, nor would this be something for urologists. Repeating this statement with reference 244 is not needed in the second paragraph and perhaps the paragraphs could be combined to be more concise.

It is not necessarily ideal that immune therapy will target CSC; not sure where this idea comes from.

The analogy of superman’s Kryptonite is perhaps inappropriate. The third paragraph could be improved by cutting the description of CSCs to one phrase.

Author Response

This is the first manuscript I have been asked to review where it is already formatted for publication. Therefore my comments will be aimed at improving the quality where possible. We thank the reviewer for their many informed comments and suggestions, which have improved our manuscript.

Introduction.

To describe UBC as more deadly in men should be reworded: the rate of death is higher in women though it kills more men in total. This is an important point given that the discussion is framed on this erroneous description of UBC as more “deadly” in men, whereas it is actually more common and less deadly. We agree and have edited all relevant passages.

Consider using actual numbers from prior years instead of estimated numbers for this time of comparison. The Siegel et al 2021 publication is cited as definitive for US cancer rates. These publications are always stated as predictions for later in the current year.

Men are at increased risk for diagnosis, but again the death rate per diagnosis is higher in women. We agree, as noted above.

References should be provided when stating a 50-70% recurrence rate; 70% is too high according to most reliable studies. Thank you for pointing this out. We have changed the text to say “up to 50%” and added two references.  

The section on prostate cancer could be more succinct and avoid certain details. For instance, a 2014 paper on the potential role of CSCs is neither recent nor seminal work in the large field of mechanism of castration resistance. We note that Schroeder et al has 86 citations, indicating that it is an influential manuscript. However, we have added more references. The term castration resistant is preferred to hormone-resistant for prostate cancer. Changes were made in reference to prostate cancer.

“Similar phenomena have been observed in other cancers that become independent of the signaling pathways that characterize cells from the same tissue of origin.” When making statements like this, it is appropriate to include references. Three citations were added.

Section 3.

While the authors suggest the common origin from the urogenital sinus suggests a reason for AR-dependence, this is speculation. Further, the prostate is considered to be mesoderm origin, which the urothelium is endodermal. We have reworded from “Therefore it may be expected  . . .” to “We speculate . . .” However, we note that under the influence of urogenital sinus mesenchyme, adult urinary bladder differentiates into prostatic tissue (Cunha GR, Fujii H, Neubauer BL, Shannon JM, Sawyer L, Reese BA: Epithelial-mesenchymal interactions in prostatic development. I. morphological observations of prostatic induction by urogenital sinus mesenchyme in epithelium of the adult rodent urinary bladder. J Cell Biol 1983, 96:1662-1670.)

In the discussion of the role of AR in patient bladder cancer, the summary statement of methodological flaws is overly harsh as while each study has limitations, they are not necessarily flawed. Similarly, the MTOPS study is not the most definitive study as indicated in the text; it is concerning that this is even mentioned given the obvious lack of power in this study. For a formalized assessment of bias of these articles and several not included, as well as a more balanced presentation the corrections should aspire to present, the authors should refer to PMID: 33132108. Thank you for the excellent review; we have cited it several times. Moreover, we have modified our discussion of the MTOPS study, which we retain because of its prospective nature.

A recent article also nicely summarizes the data on AR expression in bladder tumors: PMID: 32676741 which present data on clinical outcomes related to AR bladder cancer expression. Thank you for your suggestion. We have cited Sanguedolce et al.

Section 5

Similarly, PCa patients treated with AR blockers may display neutropenia [128]. This is an off-target effect related to flutamide and not AR antagonists. Multiple other AR antagonists do not cause neutropenia. Thank you for bringing this to our attention. We eliminated the sentence about AR blockers and neutropenia.

Section 6

It is highly unusual for a review article to switch in the middle to cite large amounts of unpublished data. The switch from narrative to first person plural also does not read well. Considering omitting this, as the review does not suffer from being to brief. Text was reduced and negative data was not specifically described.

Section 7,8

Like much of the article, these sections are very detailed. The density of material is high, making it difficult to read. The authors may consider to clearly explain a few concepts, while referring others with less explanation to make it more readable. Several sentences were eliminated or condensed.

Section 11

It might be better to rephrase the last phrase of the first paragraph since this is highly speculative. Perhaps something along the lines: “this suggests the potential for NK-cell based therapy”; it is not just a confirmatory study needed, nor would this be something for urologists. Repeating this statement with reference 244 is not needed in the second paragraph and perhaps the paragraphs could be combined to be more concise. The final sentence in paragraph 1 was modified as suggested. The final sentence in paragraph 2 was eliminated.

It is not necessarily ideal that immune therapy will target CSC; not sure where this idea comes from. We no longer say that targeting CSC will be the ideal immunotherapy.

The analogy of superman’s Kryptonite is perhaps inappropriate. The third paragraph could be improved by cutting the description of CSCs to one phrase. Kryptonite was removed and the CSC description shortened.

Reviewer 2 Report

I read this article with great interest and would like to compliment the authors for their efforts in producing their results. The study was intended to show what the role of androgenic cities in the development of bladder cancer. However, the article is not placed in a high priority neighborhood due to different methodological and writing limitations that characterize its quality. PROSPERO in order to obtain a repeatability of the result. There is a lack of graphics that can briefly illustrate the results and suggestions they have obtained to the reader. Furthermore, the tables of the included studies and the characteristics of the same are missing. Among the many weaknesses of this article I have noticed that other cancers such as breast cancer, prostate cancer are also discussed, with notable and long sections that distract the patient. reader and above all come out of the central topic of their work.
In summary, what emerges from this work is not very clear and, above all, no results are reported that can propose future research from both a translational and a clinical point of view.

Author Response

The current manuscript is not a systematic review. 

Reviewer 3 Report

Thank you for this interesting manuscript reviewing gender differences in urothelial bladder cancer.

The authors give a very detailed insight on possible basic scientific mechanisms of gender specific disparity of tumor development with special regard to the effect of natural killer lymphocyte immunity.

Literature is reviewed very extensive.

Author Response

We thank the reviewer for their generous statements. 

Round 2

Reviewer 2 Report

The authors have not improved the manuscript according to my previous comments. 

Author Response

We are disappointed that Reviewer #2 had such a negative reaction to the manuscript. The manuscript was not based on a systemic review and did not follow those guidelines. However, we did comprehensively review the literature.

Clearly Reviewer #2 is hostile to the manuscript and it seems improbable that we could revise the manuscript to meet this Reviewer’s objections.